# An Experimental and Numerical Study of CO$_2$–Brine-Synthetic Sandstone Interactions under High-Pressure (P)–Temperature (T) Reservoir Conditions

**Zhichao Yu** [1,2]**, Siyu Yang** [1,3,]***, Keyu Liu** [4]**, Qingong Zhuo** [1,2] **and Leilei Yang** [5]

1 PetroChina Exploration and Development Research Institute, Beijing 100083, China
2 Key Laboratory of Basin Structure and Hydrocarbon Accumulation, CNPC, Beijing 100083, China
3 Department of Middle East E & P, CNPC, Beijing 100083, China
4 School of Geosciences, China University of Petroleum, Qingdao 266580, China
5 Enhanced Oil Recovery Research Institute, China University of Petroleum, Beijing 100083, China
* Correspondence: yangsiy@petrochina.com.cn; Tel.: +86-10-8359-2410; Fax: +86-10-8359-2410

**Abstract:** The interaction between CO$_2$ and rock during the process of CO$_2$ capture and storage was investigated via reactions of CO$_2$, formation water, and synthetic sandstone cores in a stainless-steel reactor under high pressure and temperature. Numerical modelling was also undertaken, with results consistent with experimental outcomes. Both methods indicate that carbonates such as calcite and dolomite readily dissolve, whereas silicates such as quartz, K-feldspar, and albite do not. Core porosity did not change significantly after CO$_2$ injection. No new minerals associated with CO$_2$ injection were observed experimentally, although some quartz and kaolinite precipitated in the numerical modelling. Mineral dissolution is the dominant reaction at the beginning of CO$_2$ injection. Results of experiments have verified the numerical outcomes, with experimentally derived kinetic parameters making the numerical modelling more reliable. The combination of experimental simulations and numerical modelling provides new insights into CO$_2$ dissolution mechanisms in high-pressure/temperature reservoirs and improves understanding of geochemical reactions in CO$_2$-brine-rock systems, with particular relevance to CO$_2$ entry of the reservoir.

**Keywords:** CO$_2$ sequestration; physical simulation; Numerical modelling; dissolution; precipitation

## 1. Introduction

Carbon dioxide emissions from fossil-fuel combustion are projected to increase from 13 Gt yr$^{-1}$ in 2010 to 20–24 Gt yr$^{-1}$ in 2050 [1]. CO$_2$ capture and storage (CCS) technologies beneficially affect the lifecycle of greenhouse gases emitted from fossil-fuel power plants [2,3], with CCS expected to account for up to 19% of global CO$_2$ emission reductions by 2050, making it the most significant technology worldwide in this area [4]. Suitable geological formations for CCS include depleted oil and gas reservoirs, un-mineable coal seams, salt caverns, and deep saline aquifers [5,6]. After CO$_2$ injection, the initial physico-chemical equilibrium between saline formation fluid and reservoir rocks can be disturbed by the triggering of reactions between CO$_2$, fluid (brine), and reservoir rock [2]. Such interactions could lead to the dissolution of carbonates, feldspars, and clay cement in the aquifers [7,8]. In the absence of dynamic forces, such mineral dissolution could increase porosity and permeability by etching new pore spaces or widening narrow pore channels, temporarily increasing injectivity [9,10]. However, while sequestration of CO$_2$ in carbonate minerals can contribute to long-term storage security [11], rapid mineral dissolution, especially of carbonates, could corrode caprocks, wellbores, and fault seals,

potentially leading to migration of $CO_2$ into overlying formations. Study of $CO_2$-fluid-rock interactions is thus crucial for us to understand the physico-chemical processes involved.

Laboratory experiments can reveal the mineralogical and chemical changes resulting from $CO_2$-brine-rock interactions, how they impact the lithological porosity and permeability of the geological sequence, and the effects on CCS potential [12–15]. However, experiments are limited to short-term effects of $CO_2$ injection, whereas CCS is a long-term geochemical issue. Numerical modelling or simulation is useful for longer-tern studies. Several reactive geochemical transport models have been developed to simulate CCS, including NUFT [16], PFLOTRAN [17], CMG-GEM [18], STOMP [19], and TOUGHREACT [20,21]. The TOUGHREACT program has been widely used in studying geological $CO_2$ sequestration [22–26]. However, simulations are less reliable without the availability of parameters derived from laboratory studies, so a combination of physical experiments and numerical simulation is the optimal choice for investigating the geochemical effects following $CO_2$ injection.

In this study, both laboratory experiments (physical simulation) and numerical modelling were used to study geochemical interactions between $CO_2$-induced fluids and reservoir rock during CCS. In the physical simulation, synthetic cores with composition consistent with geological samples were used to avoid interference from other geological factors such as sedimentary processes and diagenesis. The numerical simulation involved the same conditions of sample compositions, temperature, pressure, and fluid composition, with the two simulation types being mutually authenticating. Both numerical and physical simulations were used to document the process of short-term geochemical interactions after $CO_2$ injection. A consistency of results would indicate the reliability of the simulations, with outcomes expected to be similar to those pertaining to actual geological conditions.

## 2. Samples and Methods

### 2.1. Sample Descriptions

Six synthetic sandstones were prepared for the physical simulation, with mineralogical compositions similar to sandstones of the Cretaceous Bashijiqike Formation ($K_1$bs) of the Kuqa Depression, Tarim Basin, and western China. In order to identify mineralogical compositions of $K_1$bs sandstones, the sandstone samples were prepared in thin sections and examined petrographically by point counting 300 to 400 points per section. In addition, these sandstones were also measured using quantitative X-ray diffraction analysis (D/max2500, Rigaku, Tokyo, Japan), which can provide quantitative mineralogical results within ±0.1 weight percentage (wt. %). The detail analysis processes can be found in Yu et al. (2012) [15]. The analytical results indicated that $K_1$bs sandstones are fine- to medium-grained lithic sandstones with particle sizes of 0.25~0.5 mm, comprising mainly quartz (average ~37.5 wt. %), plagioclase (~20.8 wt. %), K-feldspar (~23.3 wt. %), calcite (~9.5 wt. %), dolomite (~7.4 wt. %), and kaolinite (~1.5 wt. %) (Table 1). According to Yu et al. (2015) [27], the $K_1$bs reservoir sandstones were at the stage of mesogenetic diagenetic phase. Then we used the fine- to medium-grained mineral powders (particle size of 0.25~0.5 mm), having the above-mentioned mineralogical compositions, to reconstruct the six synthetic cores under the condition of mesogenetic diagenesis.

**Table 1.** The mineral composition of synthetic core samples.

| Mineral Types | Quartz | K-Feldspar | Albite | Calcite | Kaolinite | Dolomite |
|---|---|---|---|---|---|---|
| Content (wt. %) | 37.5 | 23.3 | 20.8 | 9.5 | 1.5 | 7.4 |

### 2.2. Physical Experimental Conditions

The experimental condition is outlined as the following: (1) 48.45 MPa back-pressure (pore fluid pressure), (2) 60 MPa confining pressure, (3) 150 °C reaction autoclave temperature (formation temperature). The injection solutions were prepared by dissolving NaCl in deionized water saturated

with $CO_2$ at 150 °C and 48.45 MPa, similar to actual $K_1bs$ conditions. The injection solutions had a salinity of 14,182 mg $L^{-1}$, approximating $K_1bs$ formation water. Here we only used the NaCl solution as the injection fluids and did not employ the imitate reservoir brines, because an amount of divalent cations, such as $Ca^{2+}$ and $Fe^{2+}$, were present in the reservoir bines. After $CO_2$ induced fluid injection into the autoclaves, some carbonates will precipitate and affect experimental results. Thus, pure NaCl solution, having a similar salinity with $K_1bs$ formation water, would be the most appropriate.

Under the experimental condition (P = 48.45 MPa and T = 150 °C), the injection solution was saturated with $CO_2$. For the solution with a salinity of 14,000 mg $L^{-1}$, the solubility of $CO_{-2}$ was 1.5451 mol/Kg, according to the $CO_{-2}$ solubility in bine of Duan and Sun (2003) [28]. During the experiment, the injected $V_{brine}$ (brine volume), and the volume of $CO_2$ injected into the cylinder was $V_{CO_2}$. Based on the equation of sate (EOS ) for gas, PV = ZnRT, where Z is the compressibility, n is the mole number of $CO_2$ ($n_{CO_2}$) in the injection solution, R is gas constant, and T is temperature, we can obtain the volume of $CO_2$ ($V_{CO_2soluble}$) dissolved in the injection solutions under the experimental condition (P = 48.45 MPa and T = 150 °C). Thus we can calculate the volume of the $CO_2$ gas cap ($V_{CO_2gc}$) in the intermediate container. The derivation is as follows:

$$V_{CO_2gc} = V_{CO_2} - V_{CO_2soluble} \tag{1}$$

$$V_{CO_2} = 1030 - V_{kerosene} - V_{brine} \tag{2}$$

$$V_{CO_2soluble} = Zn_{CO_2}RT/P \tag{3}$$

Based on the above, it is possible to calculate the volume of $CO_2$ in the gas cap of the intermediate container, which was ca.190.56 mL. Therefore the brine was $CO_2$ saturated throughout the experiments.

## 2.3. Experimental Apparatus

The physical simulation experiment was conducted at the Key Laboratory of Basin Structure and Hydrocarbon Accumulation of the China National Petroleum Corporation, Beijing, China. A reservoir diagenesis modelling system with six identical reaction autoclaves was employed (Figure 1). The system includes six modules: heating furnace, pressure system, fluid-injection system, sampling system, control panel, and auxiliary system. In addition, a corrosion-resistant HP/HT CFR-50-100 cylinder (1030 mL) from TEMC, USA was used as an intermediate container for storing the $CO_2$-bearing experimental solution. The six reaction autoclaves (Huaxing Company, Nan tong, Jiangxi Province, China) have a working pressure of 165 MPa and temperature of 300 °C. The pressure and fluid injection systems are controlled by the injection syringe pump and a back-pressure regulator. The 100DX syringe pump (Teledyne ISCO, Lincoln NE, USA) was used to control the fluid injection system, which consists of two separate systems (A and B), each of which has a capacity of 103 mL (Figure 1). It is capable of injecting at rates of 0.001~60 mL/min, with a precision of 0.5% of set point. The pump can handle pressure from 0.1 to 68.97 MPa. The advantage of this pump is its capability of continuous injection of any fluids including supercritical $CO_2$. The pore-fluid pressure was controlled by the back-pressure regulator (DBRP-005, Honeywell, USA), which has a high precision and operating pressure range up to 51.72 MPa. All experimental parameters including the injection pressure, pore fluid pressure, and temperature were monitored.

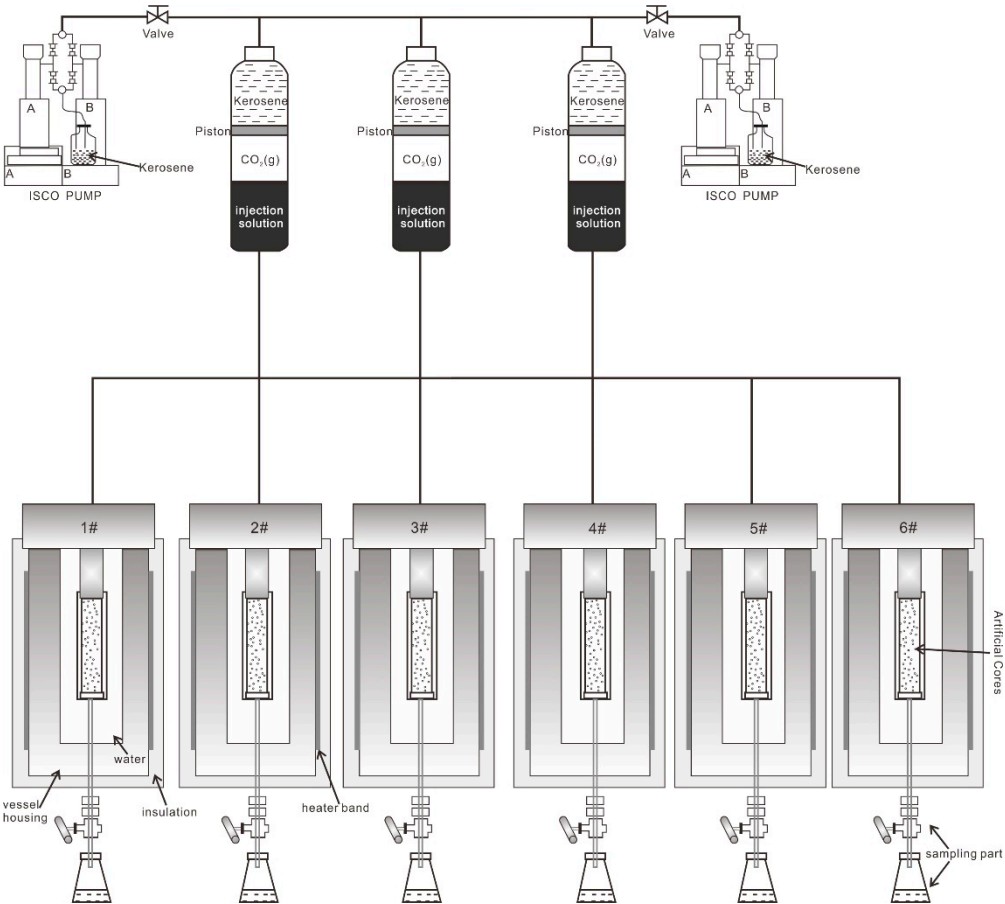

**Figure 1.** Schematic diagram of $CO_2$-formation water-rock physical experiment.

## 2.4. Physical Simulation Workflow and Analysis

The experiment was undertaken in two steps: preparation of the synthetic core, and geochemical reaction between the core and $CO_2$ fluids. During the first step, the selected mineral powder (particle size 0.25~0.5 mm) was blended with distilled water and placed in six columnar autoclaves (diameter 3.0 cm, length 11 cm, volume 77.7 $cm^3$). The six core samples (# 1 to # 6) were used in the experiment over 5 days under P/T conditions equivalent to mesogenetic diagenesis (Figure 2). The syringe-pump injection system injected synthetic formation water saturated with $CO_2$ into the six synthetic core pores at 150 °C and 48.45 MPa, after which temperature and pressure were kept constant for 4 d (# 1), 7 d (# 3), 10 d (# 4), 13 d (# 5), and 16 d (# 6), while # 2 was used as a blank.

During the experiment, the temperature and pressure of each autoclave were monitored automatically by the control system. After reaction, core and fluid samples were analyzed for ion contents, mineralogical changes, and porosity. The producing fluid was measured for its pH values using an Orion4 STAR acidity meter from Thermo within 6 h of each sampling. The ionic compositions of the water were analyzed after being spiked with 1 mol/L HCl in order to avoid carbonates precipitation, and measured using an OPTIMA 7300DX ICP-OES (Inductively Coupled Plasma–Optical Emission Spectrometry) with an analytical precision of $10^{-3}$~$10^{-9}$. Mineralogical changes were examined using a JSM6700F scanning electron microscope from JEOL with EDS (Energy Dispersive Spectrometer) from INCA software (Oxford Company, Oxford, England). The porosity changes were analyzed using visual porosity estimation, which is an image analysis technique. Firstly, core samples were impregnated with blue epoxy and then polished and made into casting thin sections. Then, combined high-resolution images of these thin sections were taken under the optical petrographic microscope; the image analysis software can delineate different types of porosity and calculate the percentages of these porosities in the thin sections with an accuracy of up to 0.01%.

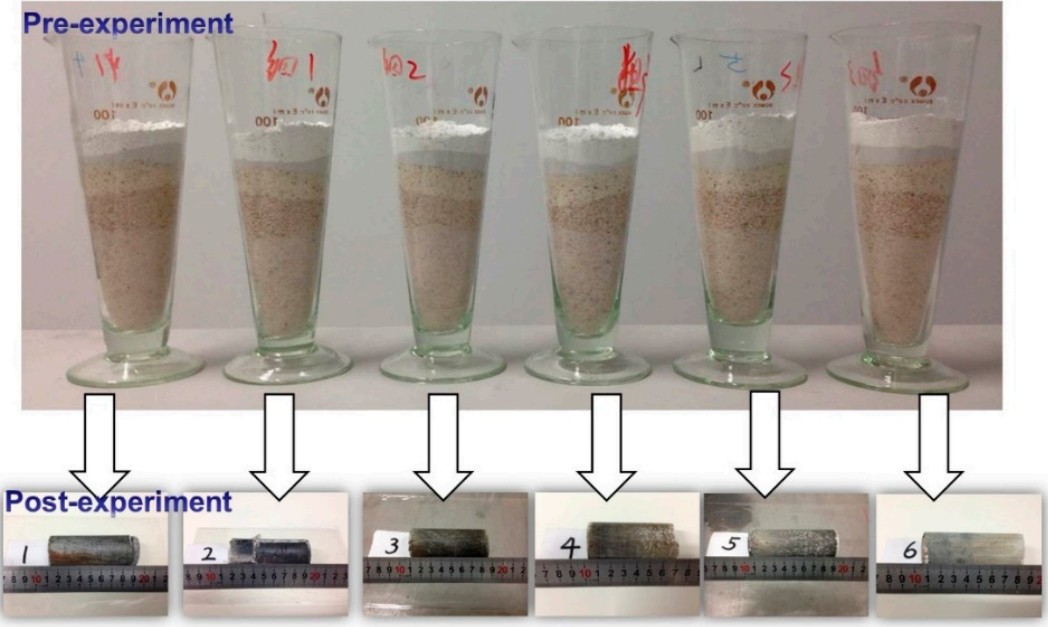

**Figure 2.** Synthetic core samples made by the physical experiment.

## 2.5. Numerical Simulation

The program TOUGHREACT was used in the numerical simulations. This program is a non-isothermal, multiphase reactive transport simulation code that was used here to simulate fluid-rock interactions [21]. The kinetic data used during the simulation are shown in Table 2.

**Table 2.** List of minerals considered and parameters for calculating the kinetic rate constants.

| Mineral | $A/(cm^2/g)$ | Geochemical Kinetic Rate Constants | | |
| --- | --- | --- | --- | --- |
| | | $K_{25}/(mol/(m^2 \cdot s))$ | $E_a/(kJ/mol)$ | $^n H^+$ |
| Quartz | 9.8 | | | |
| Kaolinite | 151.6 | $4.9 \times 10^{-12}$ | 65.9 | 0.8 |
| Illite | 151.6 | $1.0 \times 10^{-11}$ | 23.6 | 0.3 |
| K-feldspar | 9.8 | $8.7 \times 10^{-11}$ | 51.7 | 0.5 |
| albite | 9.8 | $6.9 \times 10^{-11}$ | 65.0 | 0.5 |
| Chlorite | 9.8 | $7.8 \times 10^{-12}$ | 88.0 | 0.5 |
| Calcite | 9.8 | $5.0 \times 10^{-1}$ | 14.4 | 1.0 |
| Dolomite | 9.8 | $6.5 \times 10^{-4}$ | 36.1 | 0.5 |
| Siderite | 9.8 | $6.5 \times 10^{-4}$ | 36.1 | 0.5 |
| Ankerite | 9.8 | $1.6 \times 10^{-4}$ | 36.1 | 0.5 |
| Dawsonite | 9.8 | $1.6 \times 10^{-4}$ | 36.1 | 0.5 |
| Magnesite | 9.8 | $4.2 \times 10^{-7}$ | 14.4 | 1.0 |
| Pyrite | 12.9 | $3.0 \times 10^{-8}$ | 56.9 | −0.5 |

Note that: (1) All rate constants are listed for dissolution; (2) A is specific surface area, $k_{25}$ is kinetic constant at 25 °C, $E_a$ is activation energy, and n is the power term (Equation (A1) in Appendix A); (3) The power terms n for acid mechanisms are with respect to $H^+$. Data from Palandri and Kharaka (2004) [29].

According to the columnar autoclaves employed by the physical simulation, three identical cubic grids with volumes of 77.7 $cm^3$ were used to construct the model (Figure 3). The upper and lower grids were used as boundary cells, while the middle grid was the objective model grid for simulating the processes of injection and sampling. The numerical model simulated six autoclave reactions, corresponding to the laboratory experiment, with the same mineralogical cores, temperature, pressure, and pore fluids. We used the simulation duration to mimic the six numbered autoclaves. The entire simulation ran for 16 days with intermittent sampling on day 0, 4, 7, 10, 13, and 16, corresponding

to the physical simulation. At the start of simulation (Day zero), the numerical model had an initial mineralogical composition and visual porosity, which corresponded to Autoclave # 2. In the same way, Day 4 corresponded to Autoclave # 1, Day 7 corresponded to Autoclave # 3, Day 10 corresponded to Autoclave # 4, Day 13 corresponded to Autoclave # 5, and Day 16 corresponded to Autoclave # 6. Accordingly, these results of different simulation duration from the numerical models can be used for comparison with the results from the physical simulations. The boundary cell here is an "inactive" element, whose thermodynamic conditions do not change at all from fluid or heat exchange with finite-size blocks (numerical model cell) in the flow domain. The boundary cell can confine geochemical interactions that only occur in the numerical model, which makes the results more reasonable.

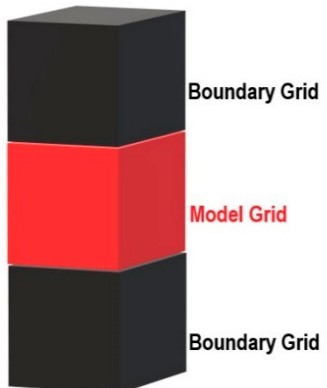

**Figure 3.** Schematic diagram of $CO_2$-formation water-rock numerical simulation.

## 3. Results

### 3.1. Changes in Fluid Chemistry

Results of physical and numerical analyses of reaction products are summarized in Table 3. Significant changes in solution chemistry were observed in both sets of experiments (Figure 4). In the physical simulation, the pH continued to increase during the 16 d of the experiment, from 5.86 to 6.44 (Figure 4). In the numerical simulation, the pH first decreased to ~2.8 within 12 d, then increased to 4.6 over the next 4 d (Figure 4).

Fluid Si and K contents show similar changes in both simulations (Figure 4), with concentrations continuing to increase with reaction time (Figure 4). Fluid Ca and Mg concentrations increased with reaction time in the physical simulation, but were more constant in the numerical simulation (Figure 4). The Al content exhibited a distinct trend (Figure 4), reaching maximum values after 7 d and 10 d for the physical and numerical simulations, respectively, and then decreasing during further reaction. Absolute value of ion concentration differed between the simulations, with the numerical simulation set generally being higher, not including pH and Al (Figure 4).

**Table 3.** Chemical composition of outlet solutions.

|  | Reaction Time (d) | pH | K | Si | Ca | Mg | Al |
|---|---|---|---|---|---|---|---|
|  |  |  | mol/L | mol/L | mol/L | mol/L | mol/L |
| **Physical Simulation** | 0 | 5.86 | 0.000000 | 0.000000 | 0.000000 | 0.000000 | 0.000000 |
|  | 4 | 5.97 | 0.000046 | 0.001000 | 0.001360 | 0.000554 | 0.000148 |
|  | 7 | 5.94 | 0.000810 | 0.001004 | 0.002500 | 0.000879 | 0.000667 |
|  | 10 | 6.01 | 0.000854 | 0.001832 | 0.003125 | 0.001079 | 0.000852 |
|  | 13 | 6.27 | 0.001987 | 0.002943 | 0.006825 | 0.002396 | 0.000500 |
|  | 16 | 6.44 | 0.002000 | 0.004500 | 0.009575 | 0.004583 | 0.000200 |

**Table 3.** *Cont.*

| | Reaction Time (d) | pH | K mol/L | Si mol/L | Ca mol/L | Mg mol/L | Al mol/L |
|---|---|---|---|---|---|---|---|
| **Numerical simulation** | 0 | 4.01 | 0.000000 | 0.000000 | 0.000000 | 0.000000 | 0.000000 |
| | 3 | 3.93 | 0.000252 | 0.000625 | 0.000631 | 0.000298 | 0.000182 |
| | 4 | 3.09 | 0.000616 | 0.001471 | 0.001344 | 0.000648 | 0.000450 |
| | 5 | 2.94 | 0.000964 | 0.002277 | 0.002119 | 0.000985 | 0.000664 |
| | 6 | 2.87 | 0.001226 | 0.002881 | 0.002733 | 0.001238 | 0.000795 |
| | 7 | 2.84 | 0.001417 | 0.003323 | 0.003178 | 0.001423 | 0.000825 |
| | 9 | 2.85 | 0.001678 | 0.003922 | 0.003846 | 0.001676 | 0.000675 |
| | 10 | 2.85 | 0.001749 | 0.004083 | 0.004035 | 0.001744 | 0.000656 |
| | 11 | 2.88 | 0.001818 | 0.004243 | 0.004192 | 0.001812 | 0.000518 |
| | 12 | 3.41 | 0.002036 | 0.004758 | 0.004568 | 0.002029 | 0.000435 |
| | 13 | 4.08 | 0.002144 | 0.005024 | 0.004732 | 0.002137 | 0.000321 |
| | 14 | 4.40 | 0.002201 | 0.005174 | 0.004819 | 0.002196 | 0.000211 |
| | 15 | 4.57 | 0.002245 | 0.005295 | 0.004891 | 0.002242 | 0.000194 |
| | 16 | 4.68 | 0.002282 | 0.005395 | 0.004953 | 0.002281 | 0.000100 |

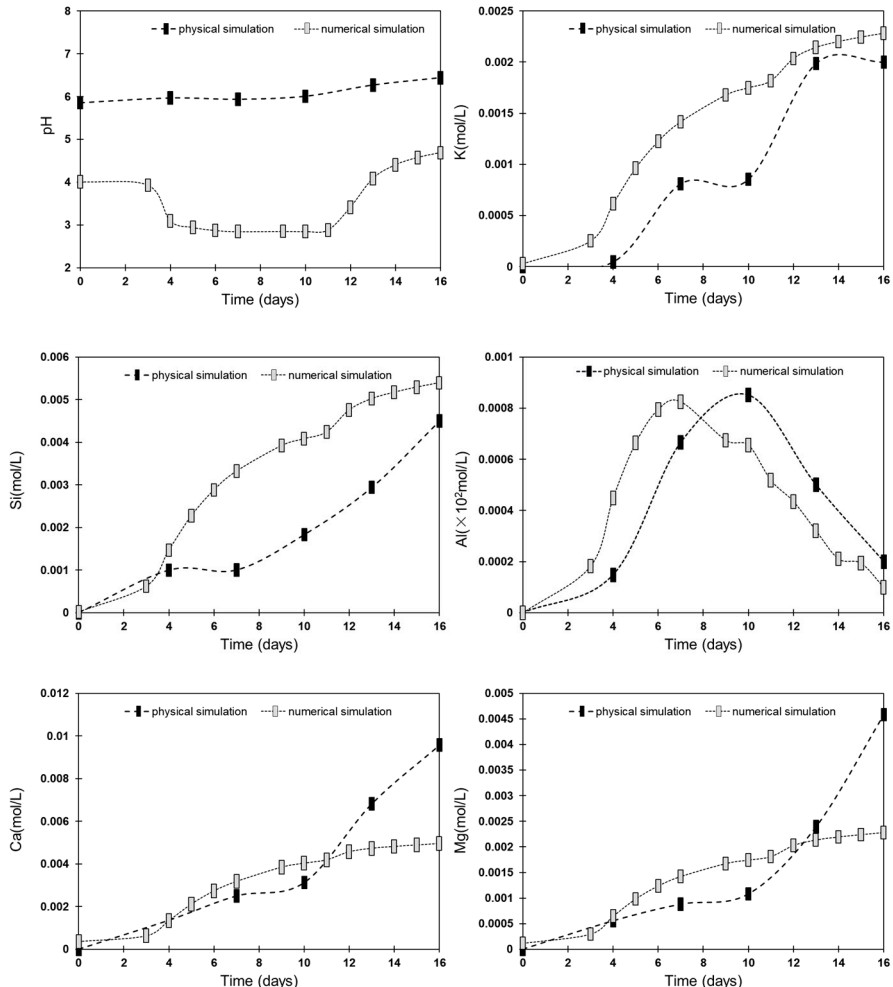

**Figure 4.** Changes of pH and typical ion concentrations over the physical and numerical simulations.

## 3.2. Changes in Mineral Morphology during the Physical Simulation

Scanning electron microscope (SEM) analyses of core samples before and after physical simulations showed that minerals such as quartz, K-feldspar, albite, and dolomite dissolved after $CO_2$ injection, with feldspar and dolomite showing pronounced dissolution and quartz weak dissolution. Before the experiment, mineral surfaces of quartz grains were generally smooth with terraced growth patterns (Figure 5A), with dissolution effects and corrosion pits being evident afterwards (Figure 5B). Initially, the albite surface was relatively flat and exhibited no obvious dissolution, but dissolution pits and fissures along cleavage surfaces were evident after the experiment (Figure 5C,D). K-feldspar was partially dissolved after the experiment, with the formation of corrosion pits (Figure 5E,F). The dissolution of K-feldspar was stronger than that of quartz and weaker than that of albite. Carbonates exhibited stronger dissolution than silicates, with entire dolomite particles being dissolved into a cloud-like phase and showing a paste-like flow structure (Figure 5G,H). Calcite was not observed after the experiment, indicating that it was completely dissolved.

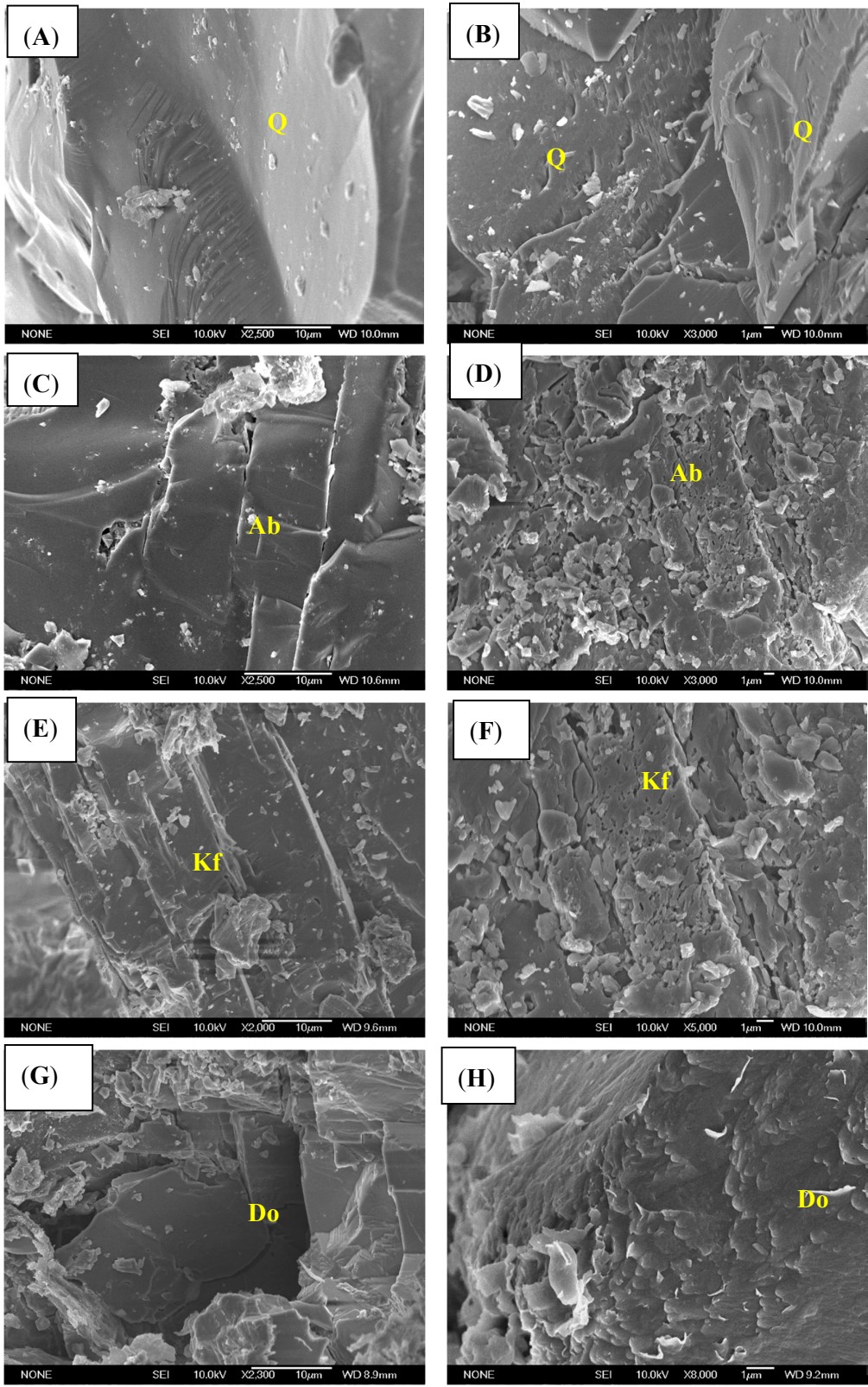

**Figure 5.** Scanning electron photomicrographs of pre-and post-experimental cores. (**A**) Quartz before the experiment; (**B**) Quartz after the experiment; (**C**) detrital albite before the experiment; (**D**). detrital albite after the experiment; (**E**) K-feldspar before the experiment; (**F**) K-feldspar after the experiment; (**G**) dolomite before the experiment; (**H**) dolomite after the experiment. Q—quartz; Ab—detrital albite; Kf—K-feldspar; Do—dolomite.

### 3.3. Changes in Porosity

Surface porosity of the synthetic cores remained relatively constant at 12.64% during the physical simulation, with no variation being observed (perhaps limited by the analytical method). Similarly, porosity changes were not evident in the numerical simulation (Figure 6), with porosity being constant up to 8 d of reaction, then increasing with carbonate dissolution to only 12.646% over the next 8 d (Figure 6).

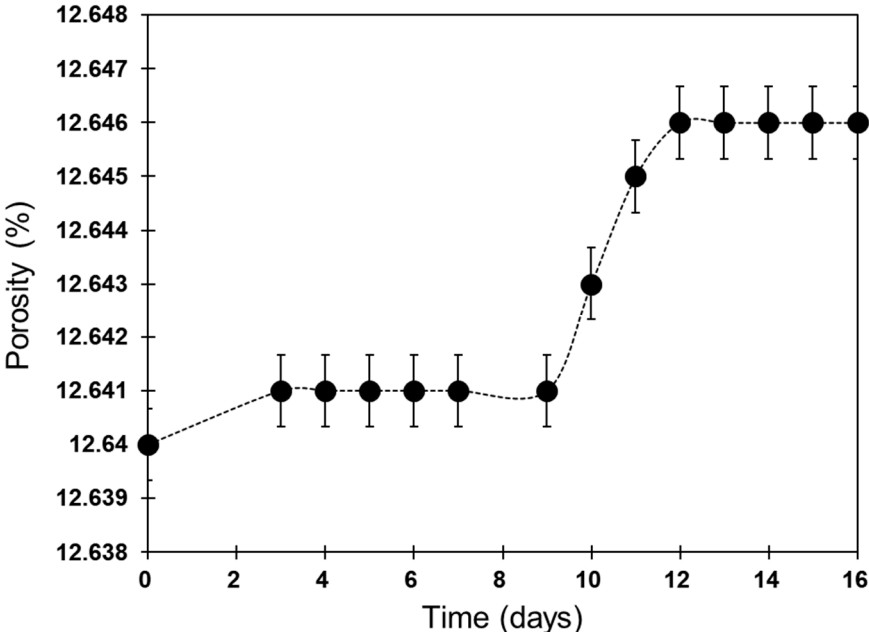

**Figure 6.** Porosity changes over time in the numerical simulation.

## 4. Discussion

### 4.1. Mineral Dissolution and Precipitation

Feldspars and carbonates are known to be easily corroded by acidic fluids during $CO_2$ injection [2,30,31], as confirmed by numerical simulations [32–34], in situ, real-time field monitoring [35,36], and natural analogies [37,38]. Changes in fluid ion contents and SEM core observations in the physical simulation confirm that feldspar and carbonate were altered by $CO_2$ injection. This is consistent with the numerical simulation, which also indicated dissolution of feldspars and carbonates (Figure 7). Both simulations indicate that Si and K, and Ca and Mg exhibit similar trends with ongoing reaction (Figure 4). Statistical analysis of Si, K, Ca, and Mg data using SPSS (Statistical Program for Social Sciences) software indicates correlation coefficients >0.5 (Table 4). Ion contents are thus likely controlled by a common reaction mechanism, as follows.

**Table 4.** Correlation coefficient matrix of the outlet solution ions.

| Correlation Matrix | K | Ca | Mg | Si | Fe | Al |
|---|---|---|---|---|---|---|
| K | 1.000 | | | | | |
| Ca | 0.943 | 1.000 | | | | |
| Mg | 0.989 | 0.979 | 1.000 | | | |
| Si | 0.921 | 0.932 | 0.955 | 1.000 | | |
| Fe | 0.877 | 0.978 | 0.938 | 0.931 | 1.000 | |
| Al | −0.035 | −0.143 | −0.126 | −0.301 | −0.330 | 1.000 |

The main mechanism controlling these reactions involves the formation of $H_2CO_3$ from dissolved $CO_2$, causing the formation water to become acidic (Equation (4)), with reducing pH. Reactions between the acidic fluid and core minerals, especially carbonates and feldspars (Equations (5)–(8), below), buffer formation-water pH, causing an increase in pH of fluid produced during the experiments [39]. This process is described by the following equations:

$$CO_2 + H_2O \rightarrow H^+ + HCO_3^- \tag{4}$$

$$CaCO_3 \text{ (calcite)} + H^+ \rightarrow Ca^{2+} + HCO_3^- \tag{5}$$

$$CaMg(CO_3)_2 \text{ (dolomite)} + 2H^+ \rightarrow Ca^{2+} + Mg^{2+} + 2HCO_3^- \tag{6}$$

$$2KAlSi_3O_8 \text{ (K-feldspar)} + 2H^+ + 9H_2O \rightarrow Al_2Si_2O_5(OH)_4 \text{ (kaolinite)} + 2K^+ + 4H_4SiO_4(aq) \tag{7}$$

$$\Delta G^0 = 18 \text{ KJ mol}^{-1}, \Delta S^0 = 73 \text{J mol}^{-1}$$

$$NaAlSi_3O_8 \text{ (albite)} + CO_2 + H_2O \rightarrow NaAlCO_3(OH)_2 \text{ (dasownite)} + 3SiO_2 \text{ (chalcedony)} \tag{8}$$

$$\Delta G^0 = -132 \text{ KJ mol}^{-1}, \Delta S^0 = -101 \text{J mol}^{-1}$$

where $\Delta G^0$ is the Gibbs free-energy change and $\Delta S^0$ is the entropy change.

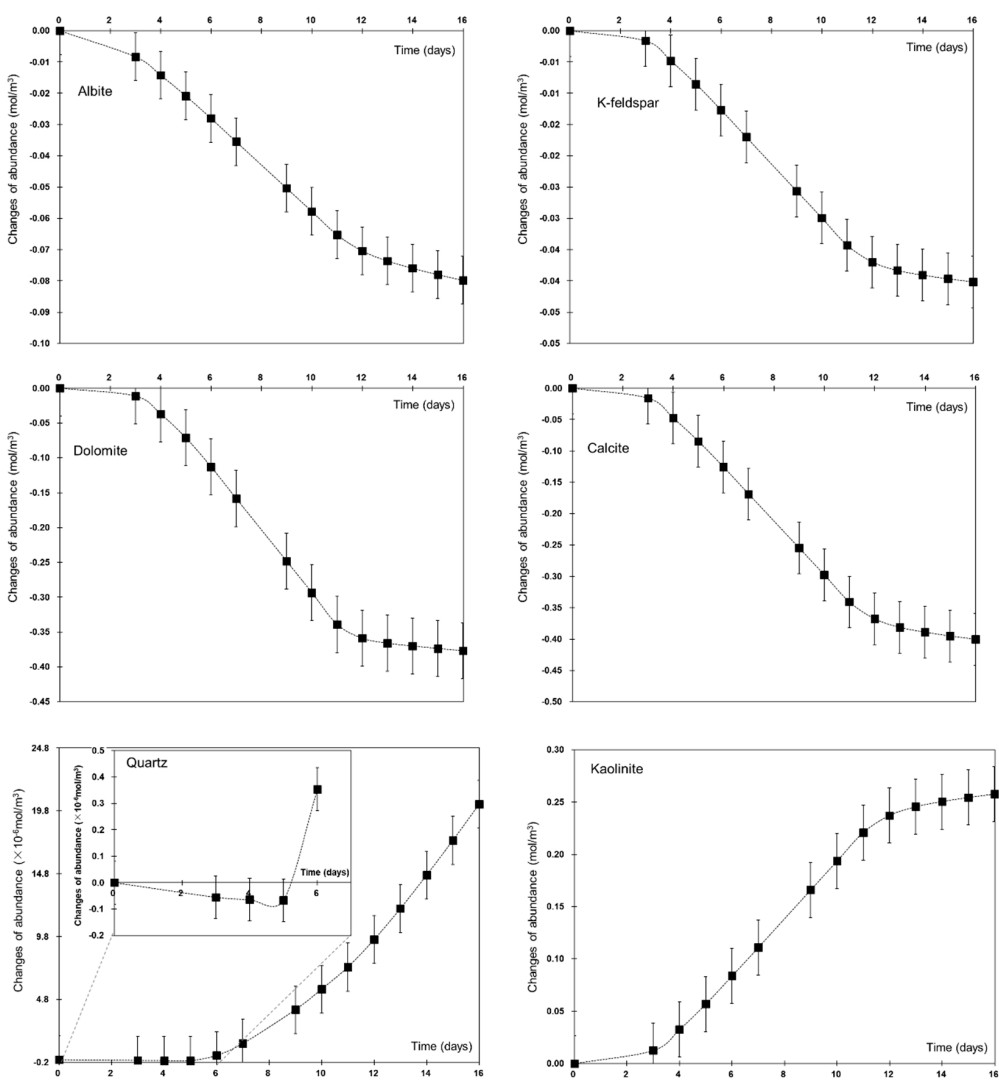

**Figure 7.** Mineral changes over time in the numerical simulation.

The degree of dissolution of albite is significantly greater (by a factor of ~2) than that of K-feldspar (Figure 7), possibly due to differences in their $\Delta G^0$ and $\Delta S^0$ values. Albite $\Delta G^0$ and $\Delta S^0$ values are both negative, with albite therefore needing little energy to dissolve, whereas K-feldspar values are positive, with more energy input needed for dissolution. The numerical simulation indicated that some kaolinite (up to 0.25 mol m$^{-3}$) and quartz (up to $19.8 \times 10^{-6}$ mol m$^{-3}$) precipitated after 4 d of reaction (Figure 7), although quartz precipitation can obviously be ignored. Equations (7) and (8) indicate that kaolinite precipitation restrains the reactions, leading to reduced K-feldspar dissolution. This is consistent with the results of other studies [15,40].

The precipitation of carbonate minerals is common during $CO_2$-induced reactions [41], and our physical and numerical results indicate that the concentrations of carbonate minerals, calcite, and dolomite all decreased significantly during reaction. In particular, dolomite was almost completely dissolved, with no carbonate minerals remaining after the experiments. This is consistent with previous experimental findings [36,37,42]. However, the numerical simulations indicate that calcite and dolomite have similar dissolution tendencies (Figure 8), whereas calcite was completely dissolved in the physical simulations. We infer that under actual geological conditions, $CO_2$ fluids react first with the most reactive minerals until they are exhausted before reacting with other minerals. In contrast, in the numerical simulations the reactions followed normal geochemical dynamic processes associated with the different minerals. Carbonate minerals did not precipitate during reaction (but produced minor amounts of kaolinite and quartz) because under the experimental conditions the reaction liquid was unsaturated with carbonates (Figure 8). Similarly, results akin to the above-mentioned calculations have also been presented by Ketzer et al. (2009) [43] and Tutolo et al. (2015) [44]. Quartz dissolution began after 5 d, and it precipitated later (Figure 8), but this reaction was very weak and is ignored here. Kaolinite was the predominant precipitated mineral (Figures 7 and 8), consistent with the results of Yu et al. (2012) [15]. However, carbonate precipitation is usually observed in $CO_2$-formation-water–rock autoclave experiments conducted in closed systems over extended periods. For example, in an experiment using Triassic Sherwood Sandstone and sea water, Pearce et al. (1996) [37] observed calcite precipitation on the sample surface in an autoclave reaction under reservoir P/T conditions after almost eight months.

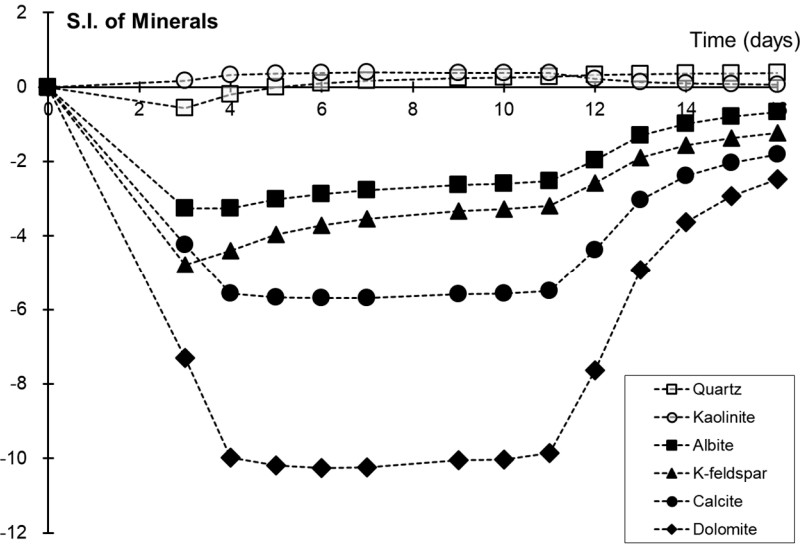

**Figure 8.** Saturation indices of carbonate minerals vs. reaction time in the numerical simulation.

Equations (4)–(6) indicate that calcite is the main reaction product, with some dolomite dissolving rapidly in the $CO_2$-saturated formation water at the beginning of the experiments. However, the silicate minerals (mainly detrital albite and K-feldspar) also gradually become unstable and start dissolving. Precipitation of clay minerals such as kaolinite occurs under acidic conditions during the reaction

process. Details of the reaction process are indicated by Equations (7) and (8). These reactions lead to a rapid increase in pH of liquid produced during the initial stage, but with pH gradually reaching a stable equilibrium value, as also observed by Bowker and Shuler (1991) [35].

*4.2. Porosity Changes*

No obvious porosity changes were observed in the synthetic core after the physical simulation, with plane porosity being constant (within measurement uncertainty) at 12.64%. This was also observed in the numerical simulation (Figure 6) where porosity only increased from 12.64% to 12.646% (Figure 6). Minor changes in mineral contents after $CO_2$ injection lead to minor changes in core porosity, as confirmed by changes in ion content in the physical simulation and other mineral changes in the numerical simulation (Figures 4 and 7). However, there were variations in porosity in the numerical simulations, where after six days of reaction the dissolution of minerals was very weak and porosity did not change noticeably, but over the following four days mineral dissolution increased with marked changes in porosity (Figures 4 and 7). Especially, a notable changes happened in porosity (Figure 6). This is due to the remarkable changes in the mineral dissolution (Figure 7). After nine days, the dissolution of feldspars and carbonates reached their peaks, indicating that the dissolution volume induced by the $CO_2$-fluid injection increased to its maximum. A large number of newly added pore spaces lead to the porosity increase. By Day 10, minerals such as kaolinite and quartz began to precipitate, with porosity becoming less variable (Figures 4, 6 and 7). Overall, porosity varied little, indicating limited dissolution and precipitation during short-term $CO_2$ injection.

The lack of reduction in porosity is common in $CO_2$-induced reactions in sandstone [7,14,40,45], with a reduction of permeability being the dominant result of short-term $CO_2$ injection. The precipitation of kaolinite, solid-phase materials, and clay particles released by the dissolution of carbonate cement may account for the non-reduction of porosity and the reduction of permeability. Shiraki and Dunn (2000) [40] considered that the precipitation of kaolinite crystals in pores is the main reason for the reduction of permeability after $CO_2$ displacement reactions, while Luquot et al. (2012) [14] considered that newly formed minerals of amorphous carbon cause the reduction in permeability. Our results also indicate that precipitation of new minerals is related to the non-reduction of porosity. In both the physical and numerical simulations, the concentration of Al increased over the first six days before decreasing over the following 10 days. In the numerical simulation, the precipitation of kaolinite occurred after six days of reaction, with this requiring large amounts of Al (Equation (7)). While minor kaolinite was precipitated during the reaction, core porosity remained almost unchanged, for two possible reasons: (1) the dissolution of minerals was very weak in short-term $CO_2$-induced reactions, with few changes occurring in feldspars and carbonates after $CO_2$ injection (<1% mol m$^{-3}$ variation); and (2) the precipitation of minerals was limited. Kaolinite content varied by a few percent, while changes in quartz content were negligible, with porosity being unchanged during such weak reactions.

The physical simulation was an autoclave experiment with the inlet connected to an injection pump (an open system), and with the outlet being a closed system opened only during sampling at the end of the experiment. The reaction system was therefore a semi-closed system. Under conditions of deep burial in semi-closed space, dissolution of carbonates rarely occurs or is very weak [46]. Regarding the volumes of water required to increase porosity through calcite or dolomite dissolution, the problem is essentially the inverse of the effect on porosity loss in limestones of calcite cementation caused by dissolved calcium carbonate from external sources [47–50]. For example, to increase the porosity of a 100 m thick limestone bed by 1%, 1 m$^3$ of calcite must be dissolved for each m$^2$ of bedding surface. For pore water that is undersaturated by 100 ppm, ~27,000 volumes of water are required to dissolve one volume of calcite. Increasing the porosity by 1% in 100 m thick limestone thus requires 27,000 m$^3$ of water per square meter of surface. Even if the limestone was underlain by 5 km of sediments in which an average porosity loss of 10% of total rock volume occurred, the pore water released from the underlying sediments would not exceed 500 m$^3$ m$^{-2}$ [46], which, in an actual geological reservoir, would not be sufficient to dissolve the carbonates. In our experiment, the autoclave volume was 77.7 cm$^3$, and it was

impossible to provide sufficient water for carbonate dissolution. However, it is certain that dissolution and precipitation are very weak at the beginning of $CO_2$ fluid-rock interactions, with our physical and numerical simulations confirming that only limited geochemical reactions, including dissolution and precipitation, occur during short-term $CO_2$ injections, with no sharp variations in core porosity or permeability. Similar results were also found by Tutolo et al. (2015), which confirmed that only very weak geochemical reactions could happen during the reaction of $CO_2$ and feldspar-rich sandstone [51]. For long-term $CO_2$ injections, however, dissolution and precipitation are the dominant geochemical processes occurring between $CO_2$-induced fluids and sandstones [51–53]. Our study of short-term geochemical interactions in a semi-closed system therefore showed no remarkable changes in the porosity of cores.

## 5. Conclusions

(1) No significant short-term $CO_2$-rock-formation-water geochemical reactions are induced by $CO_2$ injection.
(2) Neither physical nor numerical simulation found significant core porosity variations after $CO_2$ injection.
(3) Minor amounts of kaolinite and quartz were precipitated during the numerical modelling but were not observed in the physical simulation.
(4) Physical and numerical simulations conducted in tandem can be used to verify each other and improve their reliability.

**Author Contributions:** Z.Y. collected and analyzed the data and wrote the original draft. S.Y. reviewed and edited the paper. K.L. conducted the English editing work of this paper. Q.Z. contributed to the constructive discussions. L.Y. contributed to the numerical simulations.

**Funding:** This research and APC were both funded by the National Hydrocarbon Accumulation, Distribution and Favorable Areas Evaluation in Foreland Thrust Belts and Complex Tectonic Zones (No. 2016ZX05003-002).

**Acknowledgments:** We also thank the Key Laboratory of Basin Structure and Hydrocarbon Accumulation for allowing us to carry out laboratory experiments and access its rock characterization facilities.

**Conflicts of Interest:** The authors declare no conflict of interest.

## Appendix A. Kinetic Rate Law for Mineral Dissolution and Precipitation

The general rate expression used in TOUGHREACT is taken from Lasaga et al. (1994) [54]:

$$r_n = \pm k_n A_n \left| 1 - \left( \frac{Q_n}{K_n} \right)^\theta \right|^\eta \tag{A1}$$

where n denotes the kinetic mineral index, positive values of $r_n$ indicate dissolution, while negative values indicate precipitation; $k_n$ is the rate constant (moles per unit mineral surface area and unit time) and is temperature dependent; $A_n$ is the specific reactive surface area per kg $H_2O$; $K_n$ is the equilibrium constant for the mineral-water reaction for the destruction of one mole of mineral n; and $Q_n$ is the reaction quotient. The parameters $\theta$ and $\eta$ must be determined from experiments. However, they are usually, but not always, set to 1.

For many minerals, the kinetic rate constant k can be summed from three mechanisms (Palandri and Kharaka, 2004) [29]:

$$k = k_{25}^{nu} \exp\left[ -\frac{E_a^{nu}}{R} \left( \frac{1}{T} - \frac{1}{298.15} \right) \right] + k_{25}^{H} \exp\left[ -\frac{E_a^{H}}{R} \left( \frac{1}{T} - \frac{1}{298.15} \right) \right] a_H^{n_H}$$
$$+ k_{25}^{OH} \exp\left[ -\frac{E_a^{OH}}{R} \left( \frac{1}{T} - \frac{1}{298.15} \right) \right] a_{OH}^{n_{OH}} \tag{A2}$$

where superscripts or subscripts nu, H, and OH indicate neutral, acidic, and alkaline mechanisms, respectively; $E_a$ is the activation energy; $k_{25}$ is the rate constant at 25 °C; R is gas constant; T is the absolute temperature; a is the activity of the species; and n is an exponent (constant).

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
