# Peer review of "An Experimental and Numerical Study of CO2–Brine-Synthetic Sandstone Interactions under High-Pressure (P)–Temperature (T) Reservoir Conditions"

_applsci, doi:10.3390/app9163354_

Round 1

Reviewer 1 Report

The manuscript deals with a topic pertinent to the scope of the Journal. The area of the research is of relevant in the field of technologies for the mitigation of CO2 emission.

The manuscript is clearly written and well organized. However, a number of issues should be carefully checked before the publication.

The Authors are kindly invited to better explain how (and to what extent) the experimental apparatus and campaign mimic the in situ conditions. The introduction deals with the problem of CO2 capture and storage by a geological formation. How the synthetic minerals used in the study resemble the properties (i.e., the particle size) of the “real” rock formation need to be better clarified.

The Authors are clearly invited to improve the paragraph 2.3. How the “numerical model simulated the six autoclave reactions” is not clearly explained. Besides, how and why “the simulation ran for 16 days, in concert with the physical simulation” is not fully explained in detail.

Moreover, in Table 2 of the same paragraph, it is not clear whether K25 values are derived from the experiments or not (i.e., they are literature data or an input dataset in THOUGHREACT). The first raw (“acid mechanism”) does not clearly explain the meaning of the column.

The role and the meaning (as well as the effects on the simulation results, if any) of both the boundary grids need to be better clarified.

Author Response

Response to Reviewer 1 Comments

Point 1: The Authors are kindly invited to better explain how (and to what extent) the experimental apparatus and campaign mimic the in situ conditions. The introduction deals with the problem of CO2 capture and storage by a geological formation. How the synthetic minerals used in the study resemble the properties (i.e., the particle size) of the “real” rock formation need to be better clarified. 

Response 1: For this comment, we have explained and added according text in the part of “2 Samples and methods” of revised MS

Point 2: The Authors are clearly invited to improve the paragraph 2.3. How the “numerical model simulated the six autoclave reactions” is not clearly explained. Besides, how and why “the simulation ran for 16 days, in concert with the physical simulation” is not fully explained in detail.

Response 2: We have improved the described process of the numerical simulation and added according content in the paragraph 2.3, which is the paragraph 2.5 in the revised MS.

Point 3: Moreover, in Table 2 of the same paragraph, it is not clear whether K25 values are derived from the experiments or not (i.e., they are literature data or an input dataset in THOUGHREACT). The first raw (“acid mechanism”) does not clearly explain the meaning of the column.

Response 3: K25 values are literature data from the paper of Palandri and Kharaka (2004), we have added this reference in the revised MS. The “acid mechanism” is changed to be “Geochemical kinetic rate constants”.

Point 4: The role and the meaning (as well as the effects on the simulation results, if any) of both the boundary grids need to be better clarified.

Response 4: For this comment, we have added according text in the part of “2.5 Numerical simulation” of the revised MS to clarify the role and meaning of the boundary grids in the process of the numerical simulations.

Reviewer 2 Report

The paper needs to include much more detail regarding the types, detection limits, and uncertainties of analyses performed.  

For example:

What is the uncertainty on fluid chemistry measurements?  Table 3 appears to include far too many significant figures.  

How did the authors identify the mineralogy of the Bashijiqike Formation, and to what accuracy?  Where did the estimates for mineral reactive surface area (Table 2) come from?  Need to provide references.  

What is the "plane porosity" (line 165,267) and how is it measured?  

The paper also needs to include much more information about the experimental setup, including how the CO2 was dissolved into the experimental solution, and why the experimental solution employed only NaCl and did not try to more closely imitate reservoir brines. 

Other comments

Line 52: FLOWTRAN has long been replaced by PFLOTRAN: 

Lichtner P. C., Hammond G. E., Lu C., Karra S., Bisht G., Andre B., Mills R. T., Kumar J. and Frederick J. M. (2019) PFLOTRAN Web page.

Line 72: should be "X-ray diffraction" not "dispersion"

Fig. 6:  Need to include a discusion of what happens after ~9 days to cause the significant change in porosity

Text is missing reference to our previous publication on reactive transport in sandstones, where much of this study's conclusions, methods, and analysis was already published:

Tutolo B. M., Luhmann A. J., Kong X. Z., Saar M. O. and Seyfried W. E. (2015) CO2 sequestration in feldspar-rich sandstone: Coupled evolution of fluid chemistry, mineral reaction rates, and hydrogeochemical properties. Geochim. Cosmochim. Acta 160, 132–154. Available at: http://dx.doi.org/10.1016/j.gca.2015.04.002.

The entire discussion around lines 193-199 should be replaced with plots of mineral saturation index as a function of time in the numerical simulations.  The dG interpretation is not useful here.

Calculations akin to those presented on line 272 were presented previously by:

Tutolo B. M., Kong X. Z., Seyfried W. E. and Saar M. O. (2015) High performance reactive transport simulations examining the effects of thermal, hydraulic, and chemical (THC) gradients on fluid injectivity at carbonate CCUS reservoir scales. Int. J. Greenh. Gas Control 39, 285–301. Available at: http://dx.doi.org/10.1016/j.ijggc.2015.05.026.

Author Response

Response to Reviewer 2 Comments

Point 1: What is the uncertainty on fluid chemistry measurements?  Table 3 appears to include far too many significant figures. 

Response 1: For this comment, we explained the analytical process in detail and gave the analytical precision in the part of “2.4 Physical simulation workflow and analysis” of the revised MS.

Point 2: How did the authors identify the mineralogy of the Bashijiqike Formation, and to what accuracy?  Where did the estimates for mineral reactive surface area (Table 2) come from?  Need to provide references.

Response 2: We have added the text in the part of “2.1 Sample descriptions” of the revised MS to explain how we identify the mineralogy of the Bashijiqike Formation and also gave the analytical precision. For the reactive surface area, we supplemented the according reference in the table 2 of revised MS.

Point 3: What is the "plane porosity" (line 165,267) and how is it measured?

Response 3: We have added according text in the part of “2.4 Physical simulation workflow and analysis” of the revised MS to explain how we measured plane porosity.

Point 4: The paper also needs to include much more information about the experimental setup, including how the CO2 was dissolved into the experimental solution, and why the experimental solution employed only NaCl and did not try to more closely imitate reservoir brines.

Response 4: We have re-organized and added according text in the second part of the revised MS to explain experimental setup in detail. And also explain why we did not use the imitate reservoir brines in the experiment in the part of “2.2 Physical experimental conditions”.

Point 5: Line 52: FLOWTRAN has long been replaced by PFLOTRAN:

Lichtner P. C., Hammond G. E., Lu C., Karra S., Bisht G., Andre B., Mills R. T., Kumar J. and Frederick J. M. (2019) PFLOTRAN Web page.

Response 5: For this comment, we have changed accordingly.

Point 6: Fig. 6:  Need to include a discusion of what happens after ~9 days to cause the significant change in porosity

Text is missing reference to our previous publication on reactive transport in sandstones, where much of this study's conclusions, methods, and analysis was already published:

Tutolo B. M., Luhmann A. J., Kong X. Z., Saar M. O. and Seyfried W. E. (2015) CO2 sequestration in feldspar-rich sandstone: Coupled evolution of fluid chemistry, mineral reaction rates, and hydrogeochemical properties. Geochim. Cosmochim. Acta 160, 132–154. Available at: http://dx.doi.org/10.1016/j.gca.2015.04.002.

Response 6: We have explained this problem in detail in the part of “4.1 Mineral dissolution and precipitation” and added according reference which the reviewer suggested.

Point 7: The entire discussion around lines 193-199 should be replaced with plots of mineral saturation index as a function of time in the numerical simulations.  The dG interpretation is not useful here.

Calculations akin to those presented on line 272 were presented previously by:

Tutolo B. M., Kong X. Z., Seyfried W. E. and Saar M. O. (2015) High performance reactive transport simulations examining the effects of thermal, hydraulic, and chemical (THC) gradients on fluid injectivity at carbonate CCUS reservoir scales. Int. J. Greenh. Gas Control 39, 285–301. Available at: http://dx.doi.org/10.1016/j.ijggc.2015.05.026.

Response 7: For this comment, we have added new Fig 8 with plots of mineral saturation index as a function of reaction time. We also added according reference.

Round 2

Reviewer 1 Report

The manuscript has been improved on the basis of the Reviewers' comments.

Reviewer 2 Report

My previous concerns with this manuscript have now been addressed.